# Trends in Norovirus Genotypes in South Korea, 2019–2024: Insights from Nationwide Dual Typing Surveillance

**DOI:** 10.3390/v17121572

**Published:** 2025-11-30

**Authors:** Minji Lee, Seung-Rye Cho, Yunhee Jo, Deog-Yong Lee, Myung-Guk Han, Sun-Whan Park

**Affiliations:** Division of Viral Diseases, Department of Diagnosis and Analysis, Korea Disease Control and Prevention Agency, Cheongju 28159, Republic of Korea

**Keywords:** norovirus, acute gastroenteritis, molecular epidemiology, RdRp–VP1 junction, surveillance

## Abstract

Noroviruses are a leading cause of acute gastroenteritis worldwide, with a particularly high burden among children under five years of age. We analyzed nationwide surveillance data from EnterNet-Korea collected between 2019 and 2024—covering both the pre- and post-COVID-19 pandemic periods—to assess norovirus detection rates and genotype distribution. Noroviruses were detected by RT-qPCR, and dual genotyping of capsid (G) and polymerase (P) types was performed by sequencing the ORF1–2 junction region. Among the 67,159 specimens tested, 8.4% (*n* = 5652) were norovirus-positive, with the highest prevalence observed in children aged 0–5 years (19.9%). In this age group, genotyping was successful in 72.4% (2633/3635) of positive cases, identifying 30 distinct genotypes (GI: 9; GII: 21). The most common strains were GII.4[P31] (38.1%), GII.4[P16] (27.1%), and GII.2[P16] (10.4%), with substantial year-to-year variation. Notably, the emergence of GII.17[P17] in late 2024 indicates shifting genotype dynamics, underscoring the need for strengthened surveillance and reconsideration of vaccine strain selection. To our knowledge, this is the first nationwide dual-typing study in Korea encompassing the COVID-19 pandemic era. These findings provide essential baseline data for integration into global surveillance systems and inform future vaccine development and public health strategies.

## 1. Introduction

Noroviruses are recognized as one of the leading causes of acute gastroenteritis across all age groups worldwide, with a particularly high incidence among children under five years of age [1,2,3,4]. Owing to their low infective dose and multiple routes of transmission, they can spread rapidly and cause outbreaks in community and healthcare settings [1,5,6]. Noroviruses are non-enveloped, single-stranded RNA viruses belonging to the family *Caliciviridae*, with a positive-sense RNA genome of approximately 7.5 kb in length [1]. The genome consists of three open reading frames (ORFs): ORF1 encodes the nonstructural proteins, ORF2 encodes the major capsid protein VP1, and ORF3 encodes the minor structural protein VP2 [1,7,8].

Currently, the noroviruses identified are classified into ten genogroups (GI–GX), among which GI, GII, GIV, GVIII, and GIX are known to infect humans [1,9]. Each genogroup is further subdivided into capsid genotypes (G types), based on the major capsid protein (VP1) gene, and polymerase genotypes (P types), based on the RNA-dependent RNA polymerase (RdRp) gene in the ORF1 region [10]. In 2019, Chhabra et al. proposed a revised classification system using globally accumulated norovirus sequence data [10]. Based on this classification, nine capsid genotypes have been identified within GI, and more than 27 have been identified within GII, while 14 P types in GI and more than 37 in GII have been reported. GII.4 has been the most predominant genotype over the past several decades, continually generating novel variants through recombination with diverse RdRp types [11]. Such recombination events are associated with antigenic variation and enhanced epidemic potential [10,12]. Consequently, the dual typing system, which designates both VP1-based genotypes and RdRp-based P types, has been adopted as the standard classification approach for elucidating the molecular epidemiology and evolutionary dynamics of noroviruses [10,13].

Molecular epidemiological surveillance of noroviruses has been systematically implemented worldwide through national and regional networks. In the United States, CaliciNet serves as the primary surveillance platform, enabling real-time collection and analysis of sequence data from norovirus cases to monitor the emergence of epidemic genotypes and recombination events [14,15]. In Europe and other regions, NoroNet is a collaborative network involving multiple public health institutes and research laboratories, facilitating comprehensive monitoring of genotype distribution and epidemic trends [16]. In the Asia-Pacific region, the NoroNet Asia-Pacific branch provides region-specific epidemiological insights through cross-border data sharing [17,18]. Collectively, these international surveillance systems play a pivotal role in the timely detection of novel variants, the formulation of response strategies, and the establishment of baseline data essential for vaccine development. In Korea, the Korea Disease Control and Prevention Agency (KDCA) operates the “Enteric pathogen surveillance Network (EnterNet-Korea)” laboratory-based surveillance system in collaboration with primary and secondary healthcare facilities, which systematically monitors the incidence of acute diarrheal diseases and the distribution of their causative agents. This program encompasses comprehensive analyses of detection trends for noroviruses, alongside 10 bacterial genera, 5 viral pathogens, and 4 protozoa associated with acute gastroenteritis.

During the COVID-19 pandemic, the widespread implementation of non-pharmaceutical interventions (NPIs), including social distancing, mask-wearing, and enhanced hand hygiene, influenced not only the transmission of respiratory viruses but also the circulation of enteric viruses such as norovirus [7,19,20,21,22]. In Korea, similar reductions were also observed for other enteric viruses such as rotavirus and adenovirus during the pandemic, supporting the notion that NPIs substantially altered the epidemiology of multiple AGE-associated pathogens.

Previous molecular epidemiological studies conducted in Korea have primarily relied on partial capsid genotyping and were often limited to short study periods or regional data. Comprehensive nationwide surveillance applying dual typing and capturing the effects of the COVID-19 pandemic has not been reported.

This study quantitatively analyzed nationwide sentinel surveillance data to assess temporal changes in norovirus detection rates and genotype distribution over a six-year period between January 2019 and December 2024, encompassing the unique public health context of the COVID-19 pandemic. Notably, unlike previous studies in Korea that primarily focused on capsid genotyping, this is the first nationwide investigation to apply dual typing, thereby providing significant academic value and critical baseline information for future vaccine development and surveillance strategies. In addition, given rapid shifts in genotype predominance and the emergence of novel variants, continuous nationwide monitoring is essential not only for academic purposes but also for guiding vaccine strain selection and strengthening preparedness against foodborne and waterborne outbreaks in Korea.

## 2. Materials & Methods

### 2.1. Stool Sample Collection

Fecal or rectal swab specimens were collected from outpatients presenting with acute gastroenteritis (AGE) symptoms between January 2019 and December 2024. AGE was defined as the occurrence of three or more episodes of watery diarrhea per day, vomiting, nausea, and/or fever (≥37.8 °C, measured orally), which are the major clinical features of water- and foodborne infectious diseases. Specimens were obtained from patients at 76 hospitals participating in EnterNet-Korea. During the study period, minor changes occurred in the number of participating medical institutions (≤10), and the Sejong Institute of Health and Environment joined this study in August 2022. Nevertheless, the specimen collection system and diagnostic protocols remained consistent throughout the study period. Laboratory testing was performed at 18 provincial and metropolitan Institutes of Health and Environment according to the national standard operating procedures. Each institute regularly participated in external quality assessments organized by the Korea Disease Control and Prevention Agency (KDCA) to ensure testing proficiency and reliability of results. All specimens were collected as part of routine national public health surveillance conducted by the KDCA. No identifiable patient information was included in the data. In accordance with institutional policy, analyses conducted for public health surveillance are not considered human-subject research; therefore, institutional review board approval and informed consent were not required for this study.

### 2.2. Norovirus Detection

For fecal specimens, 1 g stool sample was suspended in 9 mL of 0.1 M phosphate-buffered saline (PBS), while 3 mL of PBS was added to rectal swab specimens. The suspensions were vortexed for 3 min to ensure thorough homogenization. Each fecal suspension was centrifuged at 12,000 rpm for 10 min at 4 °C, and the supernatant was collected for nucleic acid extraction. Viral nucleic acids were extracted using the NucleoMag Viral RNA/DNA Isolation Kit (Macherey-Nagel, Düren, 52355, Germany) according to the manufacturer’s instructions. Noroviruses were detected quantitative real-time reverse transcription PCR (qRT-PCR) based on the national standard method. qRT-PCR detection was performed with a commercial TaqMan assay (PowerChek™ Norovirus GI/GII Multiplex Real-Time PCR Kit, Kogene, Korea; cat. no. IR1200F) on an ABI 7500 Real-Time PCR System. Cycling: 50 °C 30 min (reverse transcription), 95 °C 10 min, then 45 cycles of 95 °C 15 s and 55 °C 60 s. Per laboratory SOP, positivity thresholds were Ct ≤ 37.2 (GI) and Ct ≤ 37.7 (GII). Specimens with Ct ≥ 36 were repeat-tested; concordant results were retained and discordant results classified as negative.

Genotyping targeted the RdRp–VP1 junction using established primers (GI: Mon432/G1SKR; GII: Mon431/G2SKR) with Sanger sequencing. Nucleotide sequences obtained in this study were submitted to NCBI GenBank (accession numbers: PX598973-PX598992, PX599239-PX599306, PX599383-PX599529, PX599585-PX599634, PX599732-PX599767, PX599773-PX599997).

### 2.3. Norovirus Genotyping

Genotyping was performed on specimens that tested positive for norovirus by qRT-PCR. For samples collected between January 2022 and December 2024, dual typing of the RNA-dependent RNA polymerase (RdRp) and capsid junction region was carried out immediately after qRT-PCR, using previously reported primer sets (GI: Mon432/G1SKR; GII: Mon431/G2SKR) [23]. For samples collected between 2019 and December 2021, only the capsid region had been analyzed at the time of detection; therefore, polymerase typing was retrospectively performed. PCR products were subjected to Sanger sequencing by a commercial sequencing service provider, and nucleotide sequences were obtained. The sequences were aligned and edited using the MEGA software package. Norovirus genotypes were determined according to the updated nomenclature system using two publicly available tools: the Calicivirus Typing Tool (https://calicivirustypingtool.cdc.gov/, accessed 30 June 2025) and the Norovirus Typing Tool provided by RIVM (https://mpf.rivm.nl/mpf/typingtool/norovirus, accessed on 30 June 2025). Sequences with insufficient reading quality or ambiguous base calls were excluded from analysis. Potential recombinant strains were screened using the RIVM Norovirus Typing Tool and confirmed by phylogenetic analysis of the partial RdRp and VP1 regions.

### 2.4. Statistical Analysis

The norovirus detection rate was calculated as the proportion of positive samples relative to the total number of specimens tested each year. Age-specific analyses were stratified into four groups (0–5 years, 6–15 years, 16–59 years, and ≥60 years), and the detection rate was determined based on the total number of samples tested within each group. For regional analysis, specimens were categorized according to the hospitals and laboratories participating in surveillance, and detection rates were calculated as the number of positive samples relative to the total tested within each region. Genotype analysis was performed on norovirus-positive samples from the 0–5 year age group. Samples lacking key patient information (e.g., age or collection date) were excluded from statistical analyses. To assess whether detection rates differed significantly across categories, a chi-square test for proportions was conducted using R (R Foundation for Statistical Computing, Vienna, Austria) and Python 3.11.5 (Python Software Foundation, Wilmington, DE, USA). A *p*-value of <0.05 was considered statistically significant for all analyses. Temporal trends in annual detection rates were assessed using the Cochran–Armitage trend test. All statistical analyses were performed with a significance threshold of *p* < 0.05.

## 3. Results

### 3.1. Norovirus Detection Rate

Norovirus was detected in 8.4% (*n* = 5652) of the samples. Among the positive cases, genogroup II (GII) accounted for 93.3% (5271/5652), while genogroup I (GI) was detected in 9.0% (*n* = 509). GI and GII were co-detected in 2.3% (*n* = 128) of the samples. Age-specific analysis revealed highest detection rate in the 0–5-year age group, with 19.9% (3635/18,291) positivity, representing 64.3% (3635/5652) of all positive cases (Table 1). Detection rates decreased progressively with increasing age. Although the ≥60 year age group contributed the largest number of tested samples (26,755/67,159), it exhibited the lowest positivity rate at 1.7% (443/26,755). Regionally, the highest norovirus detection rates were observed in Gwangju (24.4%), Seoul (15.7%), and Daejeon (10.6%), whereas the lowest rates were reported in Chungbuk (2.1%), Gyeongnam (2.8%), and Jeju (2.8%). Detailed regional detection data are provided in Appendix A. In particular, southwestern provinces such as Gwangju consistently showed the highest detection rates across all years, whereas provinces including Chungbuk, Gyeongnam, and Jeju maintained rates below 3%, suggesting potential geographic heterogeneity in norovirus transmission dynamics. Although genotype analysis focused on the 0–5 year age group, preliminary sequencing of a subset of older patients indicated similar predominance of GII.4 variants but with reduced genotype diversity, suggesting possible age-related differences in strain circulation.

Analysis of norovirus detection rates by year revealed some fluctuations over the study period. In 2019, the detection rate was 8.0% (819/10,215), which decreased significantly to 5.9% (556/9380) in 2020, coinciding with the onset of the COVID-19 pandemic (*p* < 0.001). The detection rate peaked at 9.6% in 2021 and subsequently remained at 9.0% in 2022, 8.1% in 2023, and 9.5% in 2024, showing a trend toward recovery to the pre-pandemic level observed in 2019. Monthly detection patterns are shown in Figure 1. The highest monthly detection rate was observed in January 2024 (25.8%), whereas the lowest was in September 2023 (0.5%). Detection rates exceeding 20% were predominantly observed in January–February, while lower rates (~5%) were noted during the summer and fall months (June–October). Notably, in 2022, detection rates began to rise sharply from May, reaching 21.5% in June. The Cochran–Armitage test confirmed a statistically significant upward trend in detection rates from 2020–2024 (*p* < 0.01), supporting the observation of a post-pandemic resurgence of norovirus.

### 3.2. Norovirus Genotyping (G and P Type)

Genotype analysis was performed on norovirus-positive samples from children aged ≤5 years. Among these samples, 72.4% (2633/3635) were successfully amplified by RT-PCR and sequenced. When analyzed by year, sequencing success rates for retrospectively analyzed samples from 2019, 2020, and 2021 were 66.6% (353/530), 55.6% (180/324), and 55.4% (353/637), respectively. In contrast, samples subjected to dual typing immediately after norovirus detection in 2022, 2023, and 2024 showed relatively higher sequencing success rates of 83.2% (668/803), 81.7% (589/721), and 79.0% (490/620), respectively. Among all successfully sequenced samples (*n* = 2633), a total of 31 genotypes were identified, including 9 GI and 22 GII genotypes. The distribution of the seven most frequent genotypes is shown in Figure 2A. Across both GI and GII genogroups, the most frequently detected genotype was GII.4[P31], accounting for 37.1% (977/2633) of sequences, followed by GII.4[P16] (26.4%, *n* = 695), GII.2[P16] (10.2%, *n* = 268), GII.3[P12] (7.9%, *n* = 208), and GII.6[P7] (5.6%, *n* = 148).

During 2019–2021, multiple genotypes were detected at relatively comparable frequencies of approximately 20%. In 2019, GII.4[P16] (29.5%), GII.2[P16] (23.2%), and GII.4[P31] (20.1%) together accounted for more than 70% of all cases. In 2020, the predominant genotype was GII.4[P31] (27.8%), followed by GII.3[P12] (21.7%), indicating a shift from the previous year. In 2021, GII.4[P31] remained predominant (23.2%), but GII.6[P7] (20.4%) emerged as the second most frequent genotype. However, in 2022, GII.4[P31] overwhelmingly dominated the entire year, accounting for 72.9% of all detections, followed by GII.4[P16] (16.2%), whereas other genotypes that had been more common in previous years, such as GII.3[P12], GII.6[P7], and GII.17[P17] declined substantially. In 2023, GII.4[P31] continued to predominate during the first half of the year but was superseded by GII.4[P16] in the second half. This predominance of GII.4[P16] persisted into the first half of 2024, during which GII.3[P12] also re-emerged. In the latter half of 2024, a notable increase in GII.7[P7], which had not been observed in previous years, was detected (*n* = 59). Subsequently, from November 2024 onward, GII.17[P17] markedly increased and became the predominant genotype. This sharp increase illustrates a clear genotype replacement event, in which GII.17[P17] rapidly displaced GII.4-associated strains within a single winter season.

A total of 67 samples from the GI group were successfully genotyped during the study period (Figure 2B). The most frequently detected genotype was GI.3[P13] (*n* = 23), with an annual average of six cases (range: 1–8), peaking at eight cases in 2021. The second most common genotype was GI.5[P4] (*n* = 10), seven cases of which were concentrated in 2021. GI.3[P10] (*n* = 9) was also identified, with most detections (*n* = 8) occurring in 2023, suggesting that this genotype exhibited a temporally restricted outbreak pattern. Additional GI genotypes, along with relatively rare GII genotypes (≤2% of total detections), are presented in Figure 2C. Although detected at low overall frequencies, these minor genotypes were temporally clustered, indicating that they may represent localized outbreaks or introductions from external sources. Their sporadic appearance underscores the need for sensitive surveillance capable of capturing low-level but epidemiologically relevant events.

## 4. Discussion

Norovirus is one of the most important viral agents causing acute gastroenteritis (AGE) worldwide. In Korea, surveillance of five major viral pathogens (norovirus, group A rotavirus, enteric adenovirus, astrovirus, and sapovirus) has consistently demonstrated the highest detection rates for norovirus [24]. In the present study, we analyzed nationwide data over a six-year period, including the COVID-19 pandemic, to investigate the epidemiological trends and genotype distribution of noroviruses, and further characterized the circulating strains using G–P dual typing. In the present study, the average detection rate of norovirus among children aged ≤ 5 years was 19.9% (range: 14.3–23.2%) during 2019–2024, which was slightly higher than the previously reported average detection rate of 15.2% (range: 11.8–18.9%) during 2013–2019 [25]. When considering all age groups, the detection rate in 2020 decreased significantly compared with 2019, coinciding with the onset of the COVID-19 pandemic in Korea (February 2020) and the implementation of strict social distancing and public health measures in April of the same year. Previous studies have also reported that the COVID-19 pandemic influenced the transmission of noroviruses [21,26,27,28,29]. With the introduction of NPIs, including social distancing, mask wearing, and the closure of schools and childcare facilities, the seasonal epidemic pattern of norovirus was altered, and a temporary sharp decline in detection rates was observed [7,19].

From 2021 onward, the detection rate gradually increased, returning to levels comparable to those observed in 2019. Notably, during the summer of 2022, the detection rate rose to substantially higher levels than the seasonal average, coinciding with the relaxation of social distancing stipulations and other public health measures as Korea entered the post-pandemic phase. This finding is consistent with those of previous reports describing atypical timing and scale of norovirus resurgences in several regions following the easing of public health restrictions [7,19,20]. Although a direct causal relationship between norovirus detection rates and COVID-19 cannot be established, our findings—together with those of earlier studies—suggest that stringent NPIs implemented during the early phase of the pandemic likely contributed to the suppression of norovirus and other water- and foodborne viral epidemics in Korea. Norovirus undergoes continuous genotype replacement, which is one of the major challenges for vaccine development. To date, no licensed vaccine is available, and questions remain regarding the duration of immunity and the extent of cross-protection among different genotypes [1,30]. The identification of predominant strains such as GII.4[P31], GII.4[P16], and the emerging GII.17[P17] provides critical evidence for prioritizing candidate antigens for vaccine development and assessing the breadth of cross-protective immunity in future vaccine trials. In this study, we performed genotyping analysis of positive samples obtained from children aged ≤5 years. This age group had the highest proportion of norovirus-positive cases, ensuring both representativeness of the dataset and practicality in terms of analytical efficiency and resource allocation. Sequencing success was lower in 2019–2021 due to retrospective typing from archived material (likely RNA degradation), whereas immediate dual typing since 2022 improved success. Although incomplete sequencing may introduce minor bias, the genotyped subset mirrored the temporal distribution of all positives, making substantial distortion of predominant genotype rankings unlikely.

According to our findings, the predominant genotypes detected in Korea were GII.4[P31] and GII.4[P16], which is consistent with globally reported circulating strains [1,7,20]. For example, in Shanghai, China, GII.4 Sydney[P31] and GII.3[P12] were the major genotypes detected between 2018 and 2021, with evidence of genotype replacement occurring over time [20]. In Brazil, multiple genotypes were detected even during the COVID-19 pandemic, and a re-emergence of GII.4 variants was reported in the post-pandemic period [7]. According to data analyzed from NoroSurv, the prevalence of the GII.17 genotype, which had previously been uncommon, increased in 2025 following the end of the COVID-19 pandemic [31]. Similarly, in Korea, we observed a marked increase in the detection of GII.17[P17] during the winter of late 2024. This replacement event highlights the dynamic nature of norovirus evolution and suggests that novel genotypes may rapidly establish dominance under favorable epidemiological conditions, underscoring the necessity of continuous real-time monitoring.

These findings highlight the importance of cross-country comparisons and long-term genomic surveillance for monitoring shifts in predominant genotypes [2,7,32]. In addition, although some genotypes showed low overall detection rates across the study period, they were concentrated in specific years, suggesting the possibility of transient or localized outbreaks. Given that several of these minor genotypes have been reported in neighboring countries during similar timeframes, their detection in Korea may also reflect cross-border transmissions, further emphasizing the importance of global data sharing. Therefore, continuous monitoring of circulation patterns is necessary, even for genotypes with low detection frequencies.

Despite these important findings, this study has several limitations that should be acknowledged. First, genotype analysis was restricted to samples from children aged ≤5 years, which may not fully represent genotype circulation across the entire population. Second, our sequencing strategy targeted only the RdRp–VP1 junction, which provides robust dual typing and enables detection of recombinant events but does not offer sufficient resolution for variant-level designation. As variant-defining substitutions in GII.4 lineages are distributed across the VP1, particularly within the P2 subdomain, full-length VP1 or whole-genome sequencing would be required to confidently distinguish Sydney_2012 and emerging descendant variants. To avoid over-interpretation, genotypes were therefore reported at the G-P typing level (e.g., GII.4[P31], GII.4[P16]), and Sydney_2012-like clustering was noted only where extended VP1 information was available. Future sequencing efforts incorporating full-length VP1 or whole genomes would allow a more detailed characterization of recombination dynamics and variant evolution. Additionally, the consistently higher detection rates in certain provinces, such as Gwangju, compared with persistently low rates in regions such as Chungbuk and Jeju, suggest geographic heterogeneity in transmission dynamics. These differences may reflect variations in population density, healthcare-seeking behavior, or environmental factors, and warrant further investigation. Because our data comes from a sentinel network with region-specific participation, differences in hospital mix and patient age structure may influence specimen volumes and positivity. Therefore, regional detection rates reported here should be interpreted with caution and may not directly mirror true community incidence.

Future studies should expand surveillance to include broader age groups and apply whole-genome sequencing to obtain a more comprehensive understanding of norovirus genetic diversity. In addition, strengthening data sharing through international surveillance platforms such as NoroSurv will be crucial for integrating national findings with global trends. Linking molecular epidemiology with clinical and immunological data will further contribute to vaccine development and public health preparedness. Strengthening national surveillance capacity, including facilitating timely data sharing with international platforms, will be critical for early warning of emerging variants and for guiding evidence-based public health interventions in Korea.

## 5. Conclusions

This study analyzed the genotype distribution and detection trends of noroviruses in Korea over the past six years, encompassing both the pre- and post-COVID-19 pandemic periods, using nationwide surveillance data and a dual typing approach. By integrating dual genotyping with long-term nationwide data, this study provides one of the most comprehensive molecular epidemiological datasets on noroviruses in Korea to date. The emergence of novel genotypes such as GII.17[P17] underscores the need to reconsider vaccine targets and highlights the importance of variant monitoring systems. These insights will be instrumental for informing vaccine candidate selection and evaluating the extent of cross-protective immunity across genotypes. Our findings contribute to integrating national epidemiological data into global surveillance trends and serve as valuable evidence for the development of future public health policies. Ongoing genomic monitoring, coupled with timely international data sharing, will be critical for early warning of emerging variants and for aligning Korea’s surveillance capacity with global networks. Future research should incorporate whole-genome sequencing, antigenic characterization, and integration with clinical severity data to better understand viral evolution and host–virus interactions, thereby supporting more effective vaccine development and preparedness strategies.

## Figures and Tables

**Figure 1 viruses-17-01572-f001:**
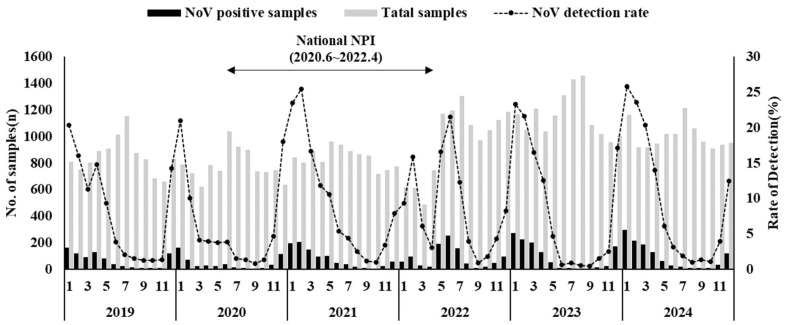
Temporal trends in norovirus detection rates in South Korea during 2019–2024. Monthly detection rates are shown for acute gastroenteritis surveillance. Non-pharmaceutical COVID-19 interventions (social distancing, mask wearing, and school closures) were implemented from June 2020 to April 2022 and are indicated on the timeline.

**Figure 2 viruses-17-01572-f002:**
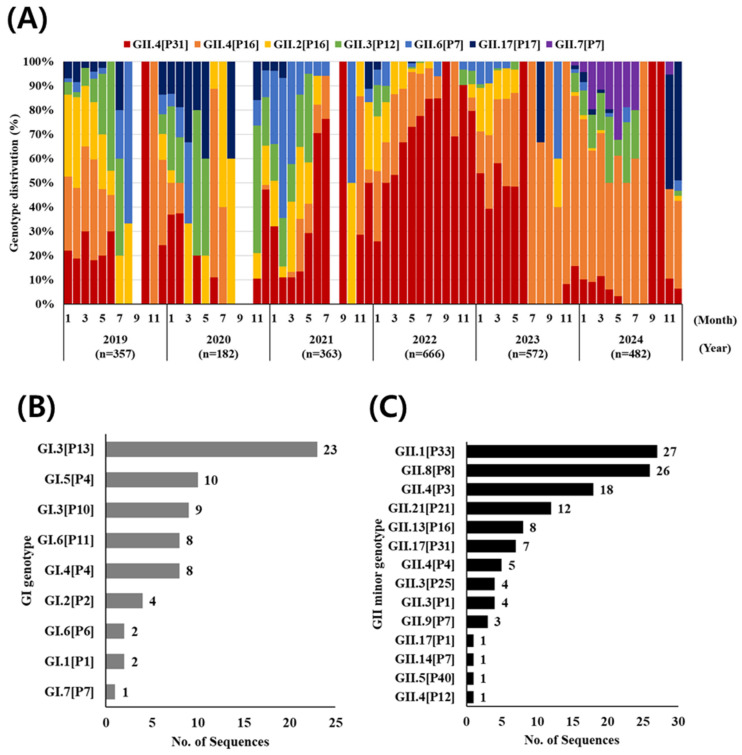
Distribution of norovirus genotypes in South Korea during 2019–2024. (**A**) Annual distribution of major genotypes (2019–2021 analyzed retrospectively). (**B**) Distribution of GI genotypes detected. (**C**) Minor GII genotypes detected at ≤2% frequency. Genotyping was based on dual analysis of the RdRp–VP1 junction.

**Table 1 viruses-17-01572-t001:** Age-wise norovirus detection rate among acute gastroenteritis cases in South Korea during 2019–2024. Data are presented as the number of positive samples over the total tested, with percentages in parentheses.

Age	No. of Samples—NoV Positive/Tested (%)						
	2019	2020	2021	2022	2023	2024	Total
0–5	530/2798	324/2258	637/3262	803/3463	721/3633	620/2877	3635/18,291
	(18.9)	(14.3)	(19.5)	(23.2)	(19.8)	(21.6)	(19.9)
6–15	98/951	74/874	169/1090	116/1083	149/1398	249/1347	855/6743
	(10.3)	(8.5)	(15.5)	(10.7)	(10.7)	(18.5)	(12.7)
16–59	120/2984	104/2360	108/2040	87/2474	147/2830	153/2682	719/15,370
	(4.0)	(4.4)	(5.3)	(3.5)	(5.2)	(5.7)	(4.7)
60<	71/3482	54/3888	52/3714	32/4535	113/6025	121/5111	443/26,755
	(2.0)	(1.4)	(1.4)	(0.7)	(1.9)	(2.4)	(1.7)
Total	819/10,215	556/9380	966/10,106	1038/11,555	1130/13,886	1143/12,017	5652/67,159
	(8.0)	(5.9)	(9.6)	(9.0)	(8.1)	(9.5)	(8.4)

## Data Availability

The original contributions presented in this study are included in the article and its Appendix A. In addition, a non-redundant, genotype-representative subset of the nucleotide sequence data generated in this study has been deposited in GenBank under accession numbers [PX598973-PX598992, PX599239-PX599306, PX599383-PX599529, PX599585-PX599634, PX599732-PX599767, PX599773-PX599997]. Further inquiries, including access to additional sequence data, can be directed to the corresponding author.

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
