# Peer review of "Trends in Norovirus Genotypes in South Korea, 2019–2024: Insights from Nationwide Dual Typing Surveillance"

_viruses, 2025, doi:10.3390/v17121572_

Round 1

Reviewer 1 Report

Comments and Suggestions for Authors

Based on the surveillance data from the National Enteric Pathogen Surveillance Network of Korea (EnterNet-Korea) between 2019 and 2024, this study systematically analyzed the detection rate and genotypic distribution characteristics of norovirus before and after the coronavirus disease 2019 (COVID-19) pandemic. From 67,159 samples, the authors detected norovirus in 8.4% of them, and identified 30 genotypes from the successfully typed samples. Specifically, in the early stage of the COVID-19 pandemic (2020), the detection rate of norovirus dropped to 5.9% due to the impact of non-pharmaceutical interventions (NPIs); subsequently, as NPIs were gradually relaxed, the detection rate gradually rebounded to the pre-pandemic level. Overall, the study provides comprehensive data, which serves as key baseline information for the integration of global norovirus surveillance networks, the selection of vaccine strains, and the formulation of targeted public health prevention and control strategies. My comments are as follows:

1.The article mentions a significant difference in the norovirus detection rate between Gwangju (24.4%) and North Chungcheong (2.1%). The reasons behind this difference are worthy of in-depth exploration. It is suggested to supplement the discussion on potential influencing factors to provide insights for the development of norovirus prevention and control programs.

2.In the methodology section, the key parameters of the PCR experiment have not been clarified; meanwhile, information such as the primer sequences, probe design, and fluorescent labeling type used in qPCR detection has not been mentioned either. It is recommended to supplement these details.

3.Viral nucleotide sequence data is the core basis of the study, and it is necessary to clearly provide relevant data (GenBank accession numbers). This is one of the key prerequisites for the article to pass the review and be accepted.

Author Response

Comment 1.

The article mentions a significant difference in the norovirus detection rate between Gwangju (24.4%) and North Chungcheong (2.1%). The reasons behind this difference are worthy of in-depth exploration. It is suggested to supplement the discussion on potential influencing factors to provide insights for the development of norovirus prevention and control programs.

Response:

We thank the reviewer for this valuable suggestion. EnterNet-Korea is a sentinel-based system; regional participation and hospital mix differ by area, which can influence specimen volumes and positivity (e.g., a higher proportion of pediatric partner hospitals in Gwangju and an age case-mix skewed toward infants/toddlers). We clarified these sentinel-system limitations and caution that regional detection rates may not directly mirror true community incidence.

Changes in manuscript:

Discussion—interpretive caution on sentinel design and regional rates (last paragraph).
“Because our data come from a sentinel network with region-specific participation, differences in hospital mix and patient age structure may influence specimen volumes and positivity. Therefore, regional detection rates reported here should be interpreted with caution and may not directly mirror true community incidence.” (lines 340–343)

Comment 2.

In the methodology section, the key parameters of the PCR experiment have not been clarified; meanwhile, information such as the primer sequences, probe design, and fluorescent labeling type used in qPCR detection has not been mentioned either. It is recommended to supplement these details.

Response:
We agree and expanded the Methods to provide the platform, cycling conditions, kit specification (including catalog number), and interpretation thresholds. As the detection kit’s primer/probe sequences are proprietary per the manufacturer, we cite the commercial kit and catalog information. For dual typing, we specify that the RdRp–VP1 junction was amplified with established primer sets and sequenced (with references).

Changes in manuscript:

Methods—“Norovirus detection”: instrument, kit (PowerChek™ Norovirus GI/GII Multiplex Real-Time PCR, Kogene), cycling profile, positivity thresholds for GI (Ct ≤ 37.2) and GII (Ct ≤ 37.7), and the repeat-test rule for high-Ct specimens. (lines 124–132)

Methods—“Norovirus genotyping”: dual typing at the RdRp–VP1 junction and references to the primer sets/tools. (adjacent lines as marked in the tracked-changes file)

Comment 3.

Viral nucleotide sequence data is the core basis of the study, and it is necessary to clearly provide relevant data (GenBank accession numbers). This is one of the key prerequisites for the article to pass the review and be accepted.

Response:

We fully agree. Sequence submissions to GenBank are in progress. We will supply accession numbers prior to acceptance and have updated the Data Availability statement accordingly.

Changes in manuscript:

Data Availability— states that nucleotide sequences will be deposited to GenBank, with accession numbers to be provided in the final version before publication. (current Data Availability paragraph)

Reviewer 2 Report

Comments and Suggestions for Authors

The specific work offers an important epidemiological analysis based on nationwide surveillance and genotyping of noroviruses circulating during a period of 5 years in South Korea. This is a significant subject in norovirus epidemiology and any short of information about the circulation and evolution of norovirus genotypes, strains and variants from any geographic region is always interesting and welcome. In this sense the specific work may be accepted for publication, providing that some major revisions are made first. Specifically, the authors do not discuss the reasons why they had low sequence success rates and which specific factors may have contributed to that. They do not also state whether these failure to sequence all available samples may have altered the reported genotype frequencies. Moreover, no supporting phylogenetic trees and recombination analyses are shown, despite reporting recombinant strains. The importance of genetic recombination in norovirus strains is not throughly discussed, in correlation with findings from other countries as well. Also, The study repeatedly refers to GII.4[P31] and GII.4[P16] as predominant but does not specify whether the genotypes belong to the GII.4 Sydney_2012 variant, or to another emergent descendant lineage. 

In my opinion, addressing these points would substantially strengthen the manuscript and enhance its scientific impact.

Author Response

Comment 1.

Specifically, the authors do not discuss the reasons why they had low sequence success rates and which specific factors may have contributed to that. They do not also state whether these failure to sequence all available samples may have altered the reported genotype frequencies.

Response:

We appreciate this point. Lower success in 2019–2021 primarily reflects retrospective typing from archived material (freeze–thaw and storage-related RNA degradation). From 2022 onwards, immediate dual typing after detection markedly improved success. While incomplete sequencing can introduce minor bias, the genotyped subset (dominated by ≤5-year-olds who contributed most positives) mirrored the temporal distribution of all positives, making substantial distortion of predominant genotype rankings unlikely.

Changes in manuscript:

Discussion—rationale for 2019–2021 lower success and bias assessment. (lines 296–300)

Results—year-specific success rates for 2019–2024 are reported as in the revised text.

Comment 2.

Moreover, no supporting phylogenetic trees and recombination analyses are shown, despite reporting recombinant strains. The importance of genetic recombination in norovirus strains is not throughly discussed, in correlation with findings from other countries as well. Also, The study repeatedly refers to GII.4[P31] and GII.4[P16] as predominant but does not specify whether the genotypes belong to the GII.4 Sydney_2012 variant, or to another emergent descendant lineage.

Response:

We agree that the text should clearly state what our junction-level data can and cannot resolve. Our sequencing strategy targeted only the RdRp–VP1 junction. This provides robust G–P typing and detects ORF1/2-junction recombination (e.g., polymerase–capsid discordance) but does not offer sufficient resolution for variant-level designation within GII.4 (e.g., Sydney_2012 descendants) across all sequences. To avoid over-interpretation, we therefore report genotypes at the G–P level and note in the Discussion that variant-level assignment would require extended VP1 or whole-genome data.

Changes in manuscript:

Discussion— scope/limits of junction-based dual typing for recombination detection and variant-level assignment; note that full VP1/WGS would be needed for distinguishing Sydney_2012 and descendant lineages. (lines 326–335)

Round 2

Reviewer 2 Report

Comments and Suggestions for Authors

I think that the authors' responses to the issues raised regarding the manuscript are acceptable. Therefore, the manuscript may be accepted for publication, with the remark that in future research regarding norovirus epidemiology, complete sequence analysis across recombination junctions and/or Whole Genome Sequencing may be pursued.